# Linking ATP and allosteric sites to achieve superadditive binding with bivalent EGFR kinase inhibitors

Florian Wittlinger[1,13], Blessing C. Ogboo[2,13], Ekaterina Shevchenko [1,3,4], Tahereh Damghani[2], Calvin D. Pham[2], Ilse K. Schaeffner[5,6], Brandon T. Oligny [2], Surbhi P. Chitnis[2], Tyler S. Beyett [5,6,12], Alexander Rasch[1], Brian Buckley[7], Daniel A. Urul[8], Tatiana Shaurova[9], Earl W. May[8], Erik M. Schaefer[8], Michael J. Eck [5,6], Pamela A. Hershberger[9], Antti Poso [1,3,4,10], Stefan A. Laufer [1,3,4,14✉] & David E. Heppner [2,9,11,14✉]

Bivalent molecules consisting of groups connected through bridging linkers often exhibit strong target binding and unique biological effects. However, developing bivalent inhibitors with the desired activity is challenging due to the dual motif architecture of these molecules and the variability that can be introduced through differing linker structures and geometries. We report a set of alternatively linked bivalent EGFR inhibitors that simultaneously occupy the ATP substrate and allosteric pockets. Crystal structures show that initial and redesigned linkers bridging a trisubstituted imidazole ATP-site inhibitor and dibenzodiazepinone allosteric-site inhibitor proved successful in spanning these sites. The re-engineered linker yielded a compound that exhibited significantly higher potency (~60 pM) against the drug-resistant EGFR L858R/T790M and L858R/T790M/C797S, which was superadditive as compared with the parent molecules. The enhanced potency is attributed to factors stemming from the linker connection to the allosteric-site group and informs strategies to engineer linkers in bivalent agent design.

[1] Department of Pharmaceutical and Medicinal Chemistry, Institute of Pharmaceutical Sciences, Eberhard Karls Universität Tübingen, Auf der Morgenstelle 8, 72076 Tübingen, Germany. [2] Department of Chemistry, University at Buffalo, The State University of New York, Buffalo, NY 14260, USA. [3] Cluster of Excellence iFIT (EXC 2180) "Image-Guided and Functionally Instructed Tumor Therapies" Eberhard Karls Universität Tübingen, 72076 Tübingen, Germany. [4] Tübingen Center for Academic Drug Discovery & Development (TüCAD2), 72076 Tübingen, Germany. [5] Department of Cancer Biology, Dana-Farber Cancer Institute, Boston, MA 02215, USA. [6] Department of Biological Chemistry and Molecular Pharmacology, Harvard Medical School, Boston, MA 02115, USA. [7] Department of Cell Stress Biology, Roswell Park Comprehensive Cancer Center, Buffalo, NY 14203, USA. [8] AssayQuant Technologies, Inc., Marlboro, MA 01752, USA. [9] Department of Pharmacology and Therapeutics, Roswell Park Comprehensive Cancer Center, Buffalo, NY 14203, USA. [10] School of Pharmacy, University of Eastern Finland, 70210 Kuopio, Finland. [11] Department of Structural Biology, University at Buffalo, The State University of New York, Buffalo, NY 14260, USA. [12] Present address: Department of Pharmacology and Chemical Biology, Emory University School of Medicine, 5119 Rollins Research Center, 1510 Clifton Rd, Atlanta, GA 30322, USA. [13] These authors contributed equally: Florian Wittlinger, Blessing C. Ogboo. [14] These authors jointly supervised this work: Stefan A. Laufer, David E. Heppner. ✉email: stefan.laufer@uni-tuebingen.de; davidhep@buffalo.edu

Molecules that simultaneously bind to distinct sites within biological targets are increasingly sought after in drug development. This binding strategy is often accomplished through bivalent (or heterobifunctional) compounds, which comprise dual functional motifs connected by a covalent linker[1,2]. Diverse and novel pharmacological strategies have emerged based on the design of such bivalent compounds that induce protein-protein neo-associations[2–9], produce protein homodimers[10–14], or associate within distinct sites of the same target[15–17]. A pivotal step in the development of these biological agents involves the optimization of the linker composition and structure, which relies on brute force exploration of synthetically accessible structural and functional motifs motivating studies that enable more efficient design strategies[18–22].

Optimization of bivalent small molecules is also a principle focus in drug development most commonly with respect to fragment-based drug discovery (FBDD) where low molecular-weight building blocks can be connected to generate high-affinity lead molecules spanning diverse binding sites[23–26]. The exploration of linked fragments is a highly attractive strategy in drug discovery due to the potential for superadditivity, such that the linked complex binds stronger than the sum of the parent fragments on their own[27–30]. Despite the simple premise, diverse efforts spanning several decades have shown that superadditivity is scarce and the majority of cases fall short of achieving the expected improvement in target affinity[20,21,28]. Additionally, case studies have offered suggestions for ideal binding properties of optimally linked compound[20,21,28,30], but little is known regarding a general structural-based strategy for swiftly optimizing fragment linkers.

The design of effective bivalent inhibitors has been inspired, in part, by the search for more effective tyrosine kinase inhibitors (TKIs)[31–33]. Kinase inhibitors most commonly bind a canonical orthosteric (ATP) binding site and more recently a set of distinct allosteric inhibitors have been reported[34–36]. The kinase domain of the epidermal growth factor receptor (EGFR) is an established drug target in non-small cell lung cancer (NSCLC) where oncogenic mutations often predict clinical responsiveness to treatment with certain TKIs[37]. Indeed, clinically-effective TKIs are often selective for EGFR-activating mutations L858R (LR) and exon19del[38], as well as drug-resistant T790M (TM) gatekeeper and C797S (CS) mutants. Promising pre-clinical results have been seen where combinations of ATP and allosteric inhibitors show synergistic tumor regression in vivo in addition to delayed acquired drug resistance[39,40]. Importantly, the EGFR allosteric inhibitor binding site is adjacent to the ATP pocket and co-binding of structurally compatible inhibitors within both sites[41,42] has been shown to enable structural changes consistent with biochemical experiments where these two inhibitor types exhibit cooperative binding[42]. The unique effects enabled by combinations of ATP and allosteric inhibitors, as well as the structural proximity of their binding sites, have led to the recent development of ATP-allosteric bivalent molecules that simultaneously occupy these sites[43,44]. Although potentially problematic when it comes to molecular weight, bivalent compounds may be especially attractive in terms of generating molecules with superior target potency and selectivity compared to co-administering two distinct inhibitors.

In this study, we synthesized a series of bivalent EGFR kinase inhibitors that simultaneously bind the ATP and allosteric sites and differ with respect to the structure of the site-bridging linker. Strikingly, we find that distinctly linked compounds exhibit considerable differences in potency where one linker exhibits superadditivity, and another is virtually inactive. Structural characterization indicates that the linker structure induces conformational differences and intermolecular interactions that provide a unique side-by-side comparison of functionally divergent linking strategies. Cocrystal structures and molecular dynamics simulations of these bivalent inhibitors enable the dissection of the specific properties of the linker that afford strong binding, which informs streamlined design strategies.

## Results

**Synthesis and Compound Design.** Due to the structural proximity of the allosteric and ATP (orthosteric) sites within the EGFR kinase, we sought to explore alternatively linked bivalent compounds that span these pockets. Starting motifs were selected and derived from established ATP-competitive inhibitors based on trisubstituted imidazole molecules (5–7)[45–49], and the mutant-selective allosteric 5,10-dihydro-11H-dibenzo[b,e][1,4]diazepin-11-one inhibitors 8 and 9 (herein denoted as "benzo" for simplicity)[50]. We synthesized a set of bivalent ATP-allosteric inhibitors bridged by an N-linked methylene (1) and C-linked amide (2–4) (Fig. 1, Supplementary Note 1, Supplementary Data 1). To combine fragments for the N-linked derivative 1 we utilized a cross-coupling focused reaction route. The motif of the allosteric site was thereby assembled by slight adjustments of previously described conditions for derivatives of the allosteric inhibitor 8[50]. The subsequent Miyaura borylation afforded the corresponding boronic acid pinacol ester, which was Suzuki coupled with the imidazole core of the orthosteric scaffold. Bromination and Suzuki coupling with the hinge-binding motif yielded 1 after acidic deprotection (Fig. 2). For the synthesis of the allosteric motif of the C-linked series we applied a Buchwald-Hartwig amination for coupling of methyl anthranilate with 3-bromo methyl anthranilate. The product was refluxed in acetic acid to obtain the methyl dibenzodiazepine-9-carboxylate by means of an intramolecular aminolysis. Saponification of the remaining ester and amide coupling of the resulting carboxylic

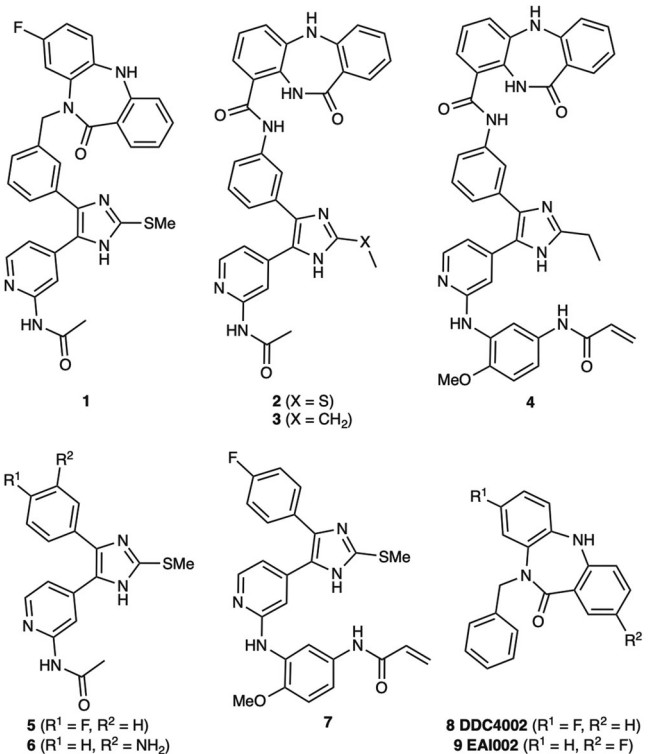

**Fig. 1 Bivalent inhibitors of EGFR developed in this study and relevant ATP and allosteric analogues for functional comparisons.** Chemical structures of bivalent ATP-allosteric inhibitors consisting of N-linked reversible (1) as well as C-linked reversible (2, 3) and covalent (4) scaffolds. Parent ATP-site imidazole reversible (5, 6) and covalent (7) inhibitors as well as dibenzodiazepinone allosteric inhibitors (8, 9).

**Fig. 2 Synthesis of the *N*-linked bivalent inhibitor (1) inspired by the parent allosteric inhibitors 8 and 9.** Reagents and conditions are as follows: i) (COCl)$_2$, DMF cat., DCM, rt; then 5-Fluoro-2-iodoaniline, Et$_3$N, DCM, rt, 38% over two steps; ii) 3-Bromobenzyl bromide, NaH (60% dispersion in mineral oil), THF, rt, 91%; iii) Fe, NH$_4$Cl, THF/MeOH/H$_2$O, 50 °C, quant.; iv) CuI, K$_2$CO$_3$, DMSO, 135 °C, 76%; v) Bis(pinacolato) diboron, KOAc, Pd(dppf)Cl$_2$, 1,4-dioxane, 90 °C, quant.; vi) 4-Bromo-2-(methylthio)-1-((2-(trimethylsilyl)ethoxy)methyl)-1*H*-imidazole, K$_3$PO$_4$ trihydrate, P(*t*-Bu)$_3$ Pd G3, 1,4-dioxane/H$_2$O, 50 °C, 70%; vii) NBS, ACN, -30 °C, 70%; viii) *N*-(4-(4,4,5,5-tetramethyl-1,3,2-dioxaborolan-2-yl) pyridin-2-yl)acetamide, K$_3$PO$_4$ trihydrate, P(*t*-Bu)$_3$ Pd G3, 1,4-dioxane/H$_2$O, 50 °C, 64%; ix) 33% TFA in DCM, rt, 62%. With adaptions from[44,50].

acid with amines of corresponding orthosteric motifs yielded *C*-linked derivatives **2, 3** and **4** after deprotection (Fig. 3, Supplementary Scheme S2).

**Biochemical Activity and Structure-Activity Relationships.** We first sought to understand the degree to which these alternatively linked motifs influence the ability to inhibit recombinant EGFR kinase activity. We carried out biochemical IC$_{50}$ value determination using HTRF-based activity assays with purified EGFR kinase domains (Table 1, Fig. 4, Supplementary Data 2). Strikingly, the *N*-linked **1** was observed to be limitedly potent against WT and all tested EGFR mutants with IC$_{50}$ values ≥ 1 µM while the *C*-linked inhibitors **2-3** show substantially lower IC$_{50}$ values of 1.2-1.5 nM for LR and 51–64 pM for LRTM and LRTMCS (Table 1, Fig. 4, Supplementary Data 2). The C797-targeting irreversible *C*-linked analogue **4** was slightly less potent as a reversible inhibitor of LRTMCS, and additional time-dependent activity measurements showed that this molecule was most effective against LR (Table 2). To put these biochemical IC$_{50}$ values into proper context, we next compared them to structurally related ATP- and allosteric-site analogues **5–9**. The ATP-site

imidazole motifs **5,6** and the original allosteric inhibitors **8,9** inhibit LRTM and LRTMCS at IC$_{50}$ values ≥ 6–10 µM and ~39–59 nM, respectively, indicating that the *C*-linked bivalent molecules are $10^3$-to-$10^6$-fold more potent over the parent motifs. The matched covalent analogue **4** inhibits LRTMCS reversibly with an IC$_{50}$ value 100-fold better than the orthosteric-only **7**, showcasing the additional reversible binding gained by interactions in the allosteric pocket in this covalent scaffold. Seeing as how the allosteric motif and linker in **2–4** is different from the *N*-linked **8,9**, we synthesized matching *C*-linked compounds of the benzo scaffolds and assayed them against LRTM (Supplementary Schemes S1 & S3 and Supplementary Fig. S1). To our surprise, **10**, which is the closest structural analogue to **2–4**, is virtually inactive and related aminothiazole-containing analogues **11** and **12** are slightly more active but with IC$_{50}$ values ≥ 10 µM (Supplementary Fig. S1). The relative inactivity of the matched benzo analogues **10–12** (≥10 µM) and ATP site analogues **5** and **6** (≥6 µM) demonstrates that the linker in the *C*-linked bivalent inhibitors **2** and **3** (51–59 pM) enables the markedly improvement in potency. Analysis of these IC$_{50}$ values allows for the estimation of the higher-limit of the linking coefficients, which are consistent with superadditivity ($E < 1$), as done previously (Supplementary Table S1)[28,30]. While the overall values are estimated to be 0.5-1.0 M$^{-1}$, they represent upper limits due to the *C*-linked allosteric **10** exhibiting virtually no activity against LRTM. The estimate of a lower-limit linking coefficient less than 1 is in line with **2** and **3** exhibiting superadditivity and confirms the *C*-linked amide as one of only a few examples of linked bivalent compounds exhibiting this behavior[20,21,28]. Corresponding calculations for lower limits of *N*-linked bivalent **1** indicate that this compound has

**Fig. 3 Synthesis of the *C*-linked amide bridged bivalent inhibitors (2–4).** Reagents and conditions are as follows: i) MeOH, H$_2$SO$_4$, rf, 56%; ii) Methyl anthranilate, Cs$_2$CO$_3$, BrettPhos Pd G3, 1,4-dioxane, rf, 73%; iii) AcOH, rf, 63%; iv) 2 N NaOH (aq), MeOH, rt, 95%; v) appropriate aniline (see supporting information), HATU, TEA, DMF, rt; then 33% TFA in DCM or MSA in DCM, rt, 30–65% over two steps. For the substitutions at position X and R see Fig. 1.

**Table 1 Biochemical EGFR IC$_{50}$ values (nanomolar) against WT and mutant EGFR kinase domains.**

| Compound | WT | LR | LRTM | LRTMCS |
|---|---|---|---|---|
| 1 | >10,000 | 1300 ± 100 | >10,000 | >10,000 |
| 2 | <10 | 1.5 ± 0.1 | 0.059 ± 0.005 | 0.064 ± 0.004 |
| 3 | <10 | 1.2 ± 0.09 | 0.051 ± 0.005 | 0.063 ± 0.005 |
| 4 | _a | _a | _a | 1.8 ± 0.3 |
| 5[45] | n.d. | n.d. | 5800 ± 300 | 6000 ± 500 |
| 6 | n.d. | n.d. | >10,000 | >10,000 |
| 7 (LN2057)[44] | _a | _a | _a | 130 ± 40 |
| 8 (DDC4002)[50] | >1000 | 690 ± 120 | 39 ± 4 | 59 ± 8 |
| 9 (EAI002)[9] | >1000 | n.d. | 52 | n.d. |

Reported IC$_{50}$ values represent best-fit values ± standard errors of a single experiment performed in triplicate.
Total enzyme concentrations WT EGFR 10 nM, LR 0.1 nM, LRTM at 0.02 nM and LRTMCS at 0.02 nM. $^a$An IC$_{50}$ value is not adequate to describe the potency of a covalent inhibitor. n.d. – "not determined."

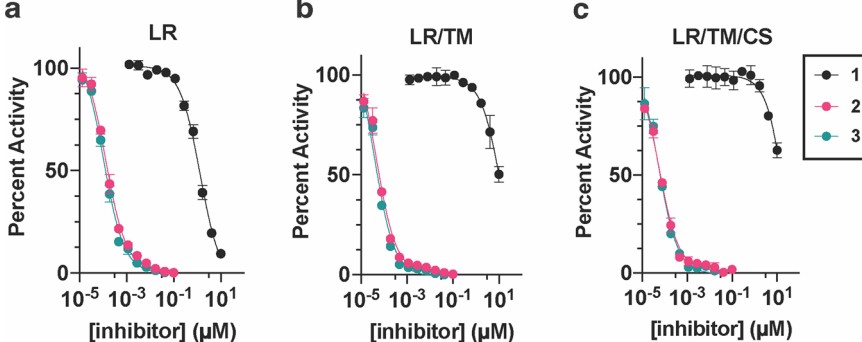

**Fig. 4 The linker structure bridging the ATP and allosteric site enables significantly altered biochemical activities of EGFR bivalent inhibitors.** HTRF-based biochemical activity dose-response curves for the reversible binding bivalent EGFR inhibitors **1–3** against (**a**) LR, **b** LRTM, and (**c**) LRTMCS mutant EGFR recombinant kinase domains. Total enzyme concentrations: LR 0.1 nM, LRTM and LRTMCS at 0.02 nM. Error bars represent standard deviation of three experimental replicates.

**Table 2 Time-dependent kinetic parameters for 4 targeting WT, LR, and LRTM EGFR recombinant proteins.**

| 4 | WT | LR | LRTM |
|---|---|---|---|
| $k_{inact}/K_I$ (M$^{-1}$s$^{-1}$) | 2500 ± 30 | 14100 ± 300 | 1070 ± 40 |
| $k_{inact}$ (min$^{-1}$) | 0.18 ± 0.004 | 0.52 ± 0.02 | 0.10 ± 0.004 |
| $K_I$ (μM) | 1.20 ± 0.04 | 0.61 ± 0.03 | 1.6 ± 0.1 |

Values obtained from global fits of progress curves ± standard errors.
Total enzyme concentrations were WT EGFR at 2.0 nM, LR at 1.0 nM, and LRTM at 2.0 nM.

distinctly higher linking coefficients (>2.0 × 10$^7$ M$^{-1}$) in line with the IC$_{50}$ activity measurements and show 10$^7$-fold differences compared to C-linked amides.

**Bivalent EGFR inhibitor binding mode characterization.** To characterize the binding modes of these bivalent inhibitors inspired by overlapping features in cocrystal structures (Fig. 5a), we determined X-ray cocrystal structures through soaking EGFR(T790M/V948R) crystals with the compounds, which reliably crystallize EGFR in the inactive (αC-helix "out") conformation (Fig. 5b, Table 3). A 2.1 Å-resolution cocrystal structure of **1** shows the imidazole and benzo groups bound within the ATP and allosteric sites, respectively, with the benzo moiety adopting an "outward" conformation (Fig. 5c, Supplementary Fig. S2A–C, Supplementary Data 3). Analogously, a 2.2 Å-resolution cocrystal structure of **2** indicates that this compound is bound identically at the ATP site as **1**, but with an opposite "inward" conformation within the allosteric pocket (Fig. 5d, e,

Supplementary Fig. S2D-F, Supplementary Data 4). Additional intermolecular interactions are observed for **2** such as H-bonding with T854 and D855 enabled by the C-linked amide, which are not possible in the N-linked methylene **1** (Fig. 5c, d). The side chain of K745, the catalytic lysine, exhibits a "swing" toward the benzo ketone in the case of **2** binding, opening a position on the imidazole, which now binds a solvent water (Fig. 5d). The conformation of the allosteric benzo moiety influences the position of the A-loop for the cocrystal structure of **1** and **2** (Fig. 5f, g, Supplementary Fig. S3). Despite the parent allosteric inhibitors **8** and **9** being best matched in terms of the N-linked **1**, the binding conformation of the C-linked **2** corresponds most closely to the allosteric inhibitor, **8** (Fig. 5h). Despite this difference in binding mode, the length of the linker is comparable between **1** and **2** (Supplementary Fig. S4). To gain a more complete understanding of the inhibitors binding and provide the insights in the activity differences, we have performed 20 μs long molecular dynamic (MD) simulations (10 replicas x 1 μs per compound) based on our cocrystal structures of **1** and **2** (Supplementary Table S2, Supplementary Figs. S5-S6). We find excellent correspondence of the ligand interaction patterns between the simulations and experimental structures including some minor variations with respect to water-mediated H-bonds with ligands not evident from the cocrystal structures (Fig. 5i, j). Additionally, computer-aided docking has provided a pose for the covalent **4** similar to what is observed for **2** with the expected orientation for covalent bond formation with C797 and consistent with our earlier structural and functional studies (GLIDE, Schrödinger, Supplementary Fig. S7)[44,49]. These cocrystal structures and MD simulations indicate these structurally related compounds with different linker structures exhibit alternative inhibitor binding modes within

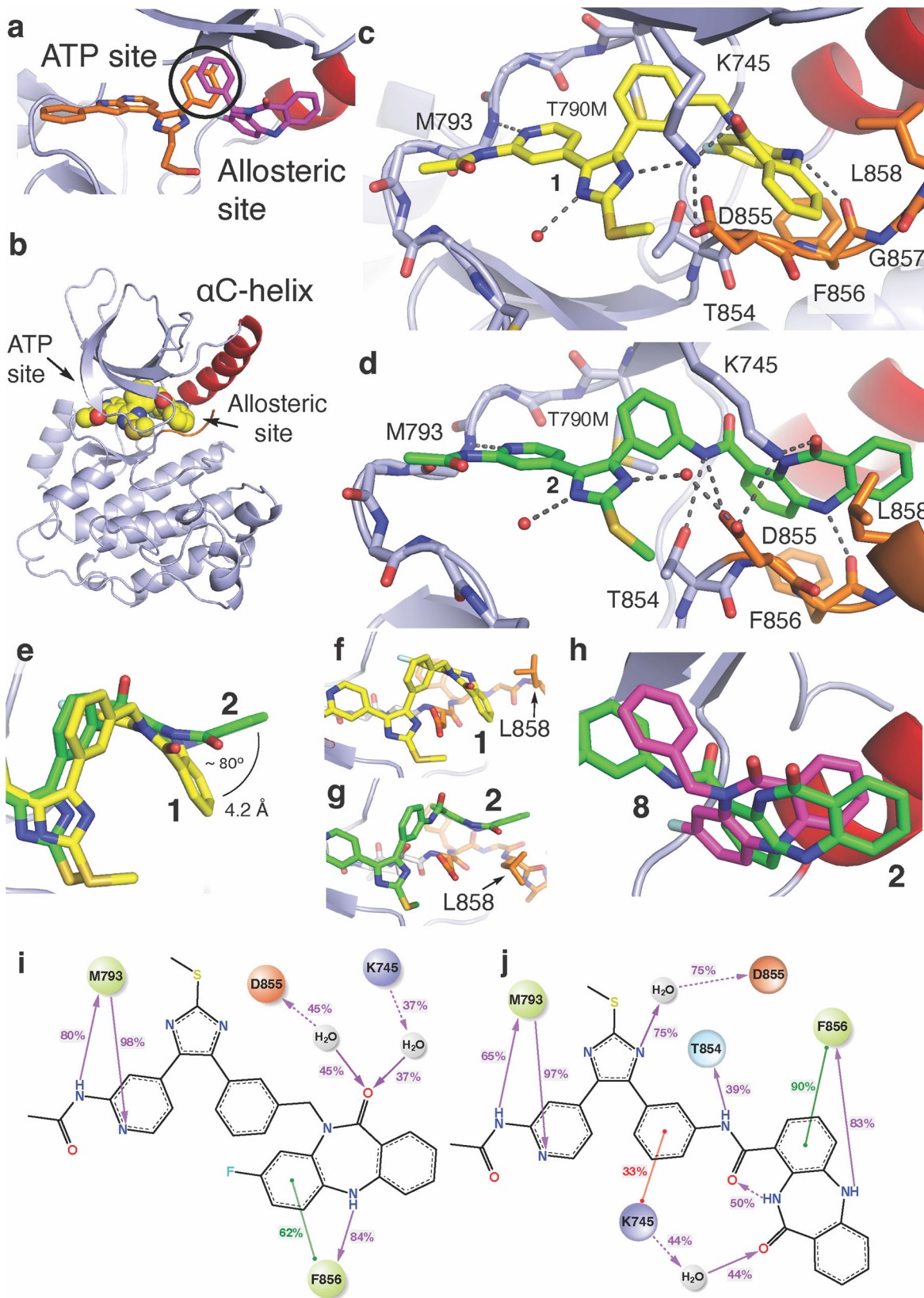

the allosteric site, side chain orientations, and intermolecular interactions.

**Factors that promote superior potency of the *C*-linked over the *N*-linked scaffold.** The wide range of potency observed for the

*C*-linked **2** and the *N*-linked **1** motivated us to more completely understand the structural basis that enables this difference in activity. Appreciating that mobility can contribute to binding, we assessed compound rigidity from crystallographic B-factors of the bound ligands **1** and **2** (Fig. 6a, b, Supplementary Fig. S8 shows changes in ligand B-factors in the context of protein backbone)[51].

**Fig. 5 Binding modes of bivalent EGFR inhibitors exhibit linker-dependent conformations within the allosteric pocket. a** The binding modes of ATP-site trisubstituted imidazole inhibitor (orange, PDB ID 6V5N) and the allosteric inhibitor DDC4002 (magenta, PDB ID 6P1D). The overlapping phenyl rings of these two compounds (bold circle) is the group at which the bivalent inhibitors are merged. **b** Overall protein-ligand structure of the EGFR(T790M/V948R) kinase domain in the αC-helix (αC) outward conformation in complex with **1** (yellow spheres, PDB ID 8FV3). **c** Active site view and binding mode of **1** (yellow) showing ATP-site binding and "outward" benzo conformation at the allosteric site (PDB ID 8FV3). **d** Active site view and binding mode of **2** (green) showing ATP-site binding and "inward" benzo conformation at the allosteric site **2** (PDB ID 8FV4) in complex with EGFR(T790M/V948R). **e** Top view overlay of **1** and **2** cocrystal structures within the allosteric pocket demonstrating the full conformational change of the benzo moiety. The angle between the benzo phenyl rings in the **2** "inward" and **1** "outward" conformation and ring-to-ring distance. The conformation of the allosteric moiety influences the positioning of L858 in (**f**) **1** (yellow) and (**g**) **2** (green). **h** View of allosteric pocket featuring an overlay of the *C*-linked bivalent **2** (green, PDB ID 8FV4) and allosteric inhibitor **8** (magenta, PDB ID 6P1D). 2D-representation of compound **1** (**i**) and compound **2** (**j**) interaction frequencies with EGFR(T790M/V948R) based on 10μs/compound MD simulations. Only interactions occurring in more than 20% of the simulation time are shown (full data is available in Supplementary Table S2, Supplementary Fig. S5). The residue and interaction color schemes are consistent for (**i**) and (**j**). Polar residues are blue, hydrophobic residues are green, negative charged residues are orange, and positive charged are purple. A green line represents π–π stacking, a red line represents the π-cations, and a purple line represents the H-bonds. A dashed line is used as an indication of side-chain interaction and a straight line of the backbone one. The interaction strength along the simulation time is shown by the percentage on the line.

**Table 3 Data collection and refinement statistics.**

|  | 1 (PDB ID 8FV3) | 2 (PDB ID 8FV4) |
|---|---|---|
| **Data collection** | | |
| Space group | P 1 2₁ 1 | P 1 2₁ 1 |
| Cell dimensions | | |
| $a, b, c$ (Å) | 72.3699, 103.224, 87.1399 | 70.6064, 100.485, 87.3183 |
| $\alpha, \beta, \gamma$ (°) | 90.00, 101.48, 90.00 | 90.00, 102.323, 90.00 |
| Resolution (Å) | 65.8 - 2.1 (2.175 - 2.1) | 60.3 - 2.2 (2.279 - 2.2) |
| $R_{merge}$ | 0.1376 (0.7708) | 0.0732 (0.4283) |
| $I / \sigma I$ | 7.05 (0.82) | 9.73 (1.04) |
| Completeness (%) | 99.92 (99.81) | 98.33 (98.73) |
| Redundancy | 6.8 (6.8) | 3.4 (3.4) |
| **Refinement** | | |
| Resolution (Å) | 65.8 - 2.1 (2.175 - 2.1) | 60.3 - 2.2 (2.279 - 2.2) |
| No. reflections | 73205 (7310) | 59494 (5975) |
| $R_{work} / R_{free}$ | 0.1872 / 0.2264 (0.2429 / 0.3178) | 0.2012 / 0.2606 (0.2861 /0.3655) |
| No. atoms | | |
| Protein | 9429 | 9307 |
| Ligand/ion | 137 | 106 |
| Water | 336 | 198 |
| *B*-factors | | |
| Average B-factor (Å²) | 35.71 | 43.72 |
| Protein | 35.67 | 43.72 |
| Ligand/ion | 36.72 | 48.37 |
| Water | 36.42 | 43.12 |
| R.m.s. deviations | | |
| Bond lengths (Å) | 0.018 | 0.009 |
| Bond angles (°) | 1.15 | 1.01 |

Values in parentheses are for highest-resolution shell.

This is made possible due to several commonalities shared between these cocrystal structures, including as they originate from the same protein, unit cells, and atomic resolutions (2.1 Å for **1** and 2.2 Å for **2**). Generally, the ATP-binding imidazole in both compounds are comparably rigid with B-factors below the structure average while a notable increase is observed for the allosteric moiety in **2** and to a much lesser extent in **1** (Fig. 6a, b). To gain deeper insight, we performed generalized Born and surface area solvation (MM-GBSA) calculations on MD trajectories using our cocrystal structures (Fig. 5c, d). These calculations provide Free energies of binding where **2** exhibits greater affinity than **1** ($\Delta\Delta G = 9.5$ kcal/mol), consistent with the difference in IC₅₀ values (Table 1), which is enabled by enhanced van der Waals and H-bonding interactions (Supplementary Table S3-4, Supplementary Fig. S9). Additionally, MM-GBSA

ligand energy calculations indicate a ~ 3.4-fold lower energy for **2** compared to **1**, implying that **2** possesses a greater degree of structural complementarity within the kinase binding sites (Supplementary Table S3, Supplementary Fig. S10). Superior binding of **2** is also aided by the "inward" benzo binding mode as this conformation is capable of full displacement of energetically unfavorable water molecules and the "outward" conformation of **1** does not allow complete displacement (Fig. 6c, Supplementary Fig. S11). Further energetic analysis indicates that the potential energy of **2** pertaining to conformation within the binding mode of **2** is ~4-fold more favorable than the corresponding conformation of **1** (Fig. 6d, e). The overall pictures obtained by the crystallographic B-factors and MD simulations indicate that the superior potency of the *C*-linked compounds is due to a variety of factors that all stem from the structure of the linker allowing for improved mobility and pocket complementarity within the allosteric site. How the structure of the linker impacts the binding mode of these compounds is best visualized in terms of torsion angles observed in the MD simulations (Fig. 6f, g, Supplementary Fig. S12). The most unique rotatable bond in **2** is the C-C bond that connects the linker amide to the benzo via the back pocket phenyl ring and allows for enhanced mobility of the group within the allosteric pocket (orange arrow Fig. 6g). This rotatable bond is the key structural element that allows for tight binding of this compound to EGFR since the other rotatable bonds in the linkers of **1** and **2** are comparably rigid and anchored to the relatively static ATP-site imidazole. These experimental and theoretical studies reveal the molecular factors that enable effective bivalent inhibitor binding, which can all be attributed to the "inward" conformation within the allosteric pocket enabled by the enhanced linker-enabled mobility of the *C*-linked scaffold.

**Cellular Activity of the Bivalent Inhibitors**. We next sought to gauge the biological activity of our *C*-linked ATP-allosteric bivalent inhibitors. The Michael acceptor-containing *C*-linked analogue **4**, designed to target C797 as done previously[44,46–48], was found most effective at suppressing LRTM phosphorylation (pY1068) as well as downstream pERK and pAKT in the human NSCLC cell line H1975 below 1 μM concentration dosed for 6 h (Fig. 7a, Supplementary Fig. S13-S14). Additional studies with H3255 (LR), H3255GR (LRTM) and HCC827 (exon19 delE746-A750) cells exhibit analogous suppression of EGFR pY1068 phosphorylation, slightly better potency in H3255 and H3255GR, cells indicating that **4** broadly targets diverse EGFR mutations (Fig. 7b, c, Supplementary Figs. S15-S16). The reversible binding **2**-**3** are also effective in H1975 cells, however to a lesser extent than **4**, while the *N*-linked **1** exhibits limited ability to suppress phosphorylation (Supplementary

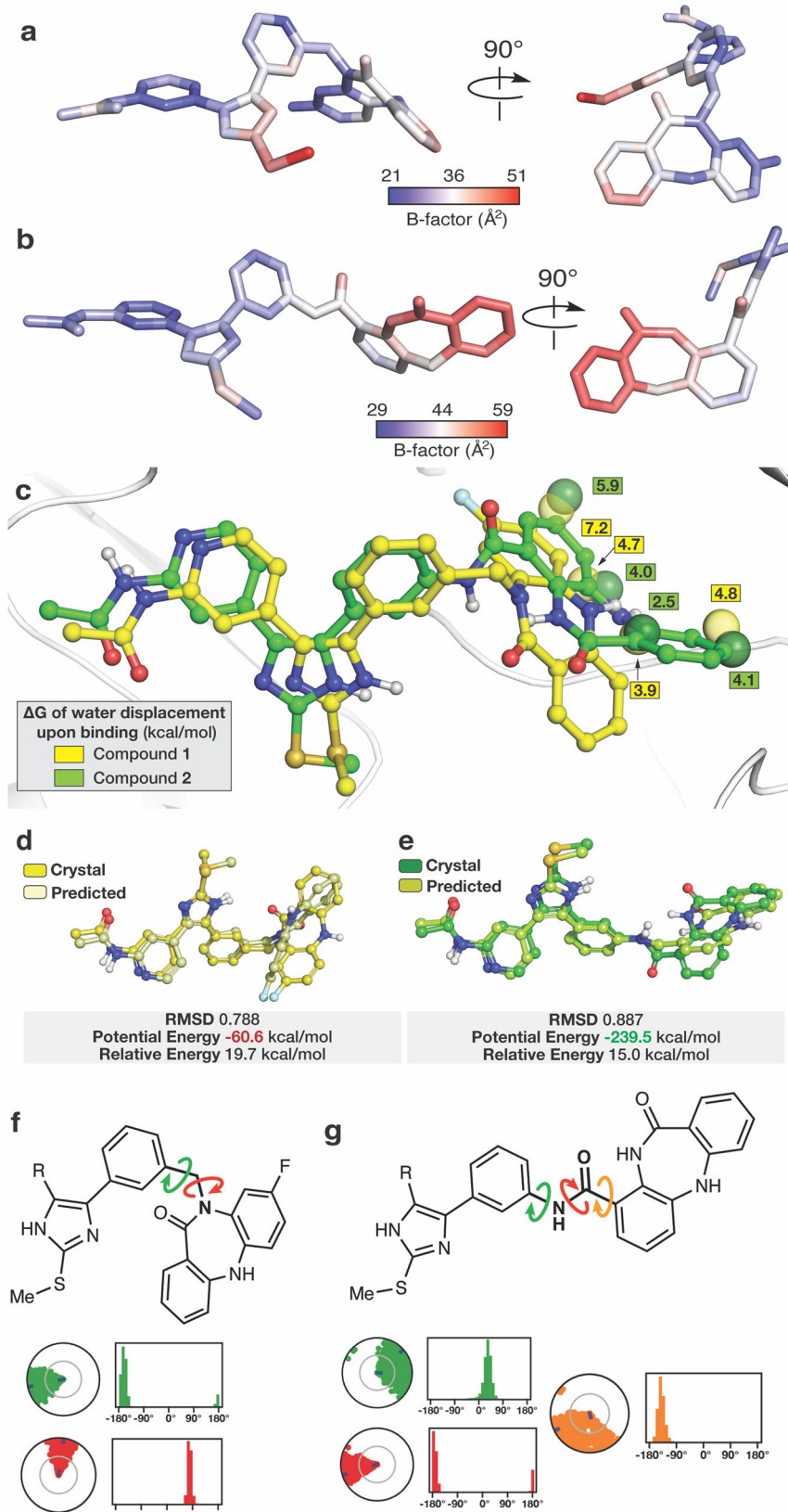

Fig. S17). We have also determined antiproliferation effects in human cancer cell lines H1975 and HCC827 treated with **4**, which show that this EGFR inhibitor is active at concentrations ~100–500 nM overall ~60-fold less potent than AZD9291 in both cell lines (Supplementary Table S5, Supplementary Fig. S18). While activity in H1975 and H3255 cells is expected based on biochemical data (Figs. 4 and 5), **4** is unexpectedly effective against HCC827 cells that harbor the prominent EGFR exon19 delE746-A750 mutation as allosteric pocket binding compounds are typically ineffective against this mutation (Fig. 7c)[52]. While the details regarding the structural basis for the inhibition of this deletion mutations are under-developed at present, recent MD simulations indicate that an "αC-

**Fig. 6 Structural basis for linker-dependent strong binding of the *C*-linked bivalent inhibitor 2.** Variations in crystallographic B-factors of the bound ligands (**a**) **1** and (**b**) **2** in complex with EGFR. The blue-to-red gradient is averaged to the overall average B-factors are 36 Å$^2$ for **1** (PDB ID 8FV3) and 44 Å$^2$ for **2** (PDB ID 8FV4). **c** WaterMap simulation of the unbound EGFR(T790M/V948R) binding pocket for compounds **1** and **2**, highlighting unfavorable hydration sites (yellow spheres for **1** and green spheres for **2**) and their relative location to bound **1** and **2**. Color-coded ΔG values reflect the compound associated with each water molecule (See Supplementary Fig. S11 for complete analysis). Conformational analysis of compound (**d**) **1** and (**e**) **2** (MacroModel). The potential energy values highlight that **2** adopts a conformation that is over 3-fold more energetically favorable in the binding site compared to **1** in the binding site. Relative potential energy indicates the energy difference to the lowest energy conformation in the predicted set and highlights a closer alignment to the ideal conformation for **2** compared to **1**. The root-mean-square deviation (RMSD) was used to justify the ligand conformation closest to the crystal structure. Ligands are displayed with ball and stick representations using yellow and green colors for carbon-atoms of **1** and **2**, respectively. Chemical structures of (**f**) **1** and (**g**) **2** denoting linker rotatable bonds and their torsion angles from MD simulations (see Supplementary Fig. S12 for complete analysis).

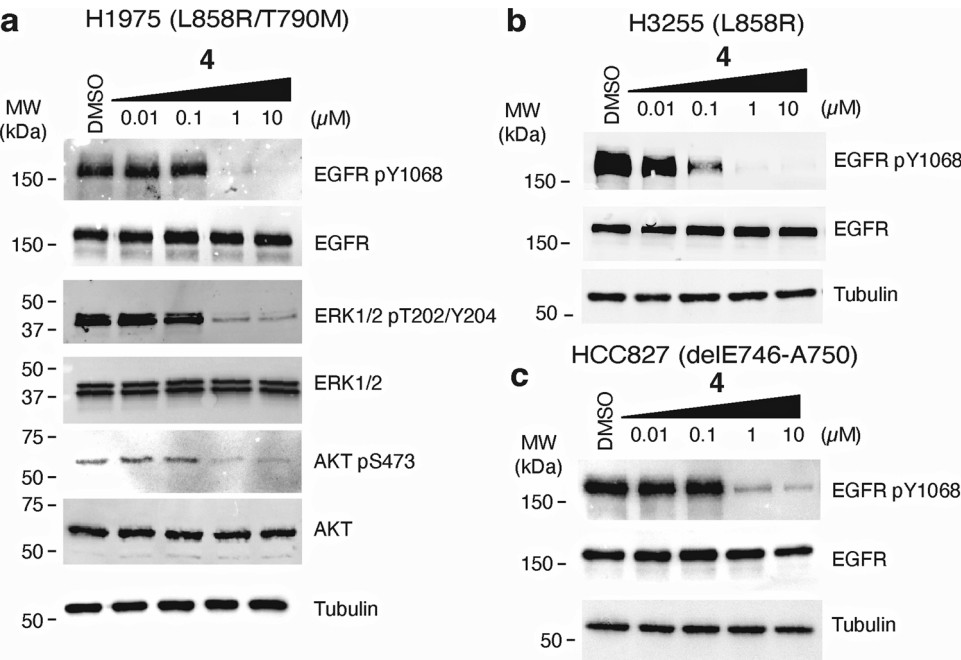

**Fig. 7 The bivalent inhibitor 4 suppresses EGFR phosphorylation across diverse human NSCLC cell lines.** Impact of **4** on EGFR phosphorylation (pY1068) in **a** H1975, **b** H3255, and **c** HCC827 cell line models. **a** Dosing of **4** in H1975 cells also diminishes phosphorylation of ERK1/2 and ATK downstream signaling kinases. Western blot experiments performed after 6-hour treatments and representative of at least three independent experiments ($N = 3$).

helix outward" conformation may be accessible allowing for the binding of ATP-allosteric bivalent molecules[53]. Additional anti-proliferative activity experiments in Ba/F3 cells are consistent with above observations, and the ~220 nM EC$_{50}$ value for LR is markedly improved compared to our earlier covalent bivalent ATP-allosteric inhibitors indicating that the benzo-derived scaffold exhibits improved cellular activity (Supplementary Table S6)[44]. Despite this advancement, our bivalent inhibitor **4** is generally less effective in TM and CS-containing cells as compared to AZD9291, most likely due to poorer membrane permeability on account of molecular weight and size. Furthermore, we confirm that **4** is selective across the kinome exhibiting a selectivity score of S(35) = 0.084 (Supplementary Table S7) and metabolically stable in liver microsome assays (Supplementary Fig. S19). These experiments indicate that the *C*-linked bivalent inhibitor **4** is capable of targeting cellular EGFR in biological context of several prevalent oncogenic activating mutants and motivates further efforts to optimize the potency and medicinal chemistry properties for translational studies.

## Discussion

Bivalent (heterobifunctional) compounds are attractive molecules in chemical biology and drug discovery for their unique properties and accessibility toward classically "undruggable" biological targets[1,2,54–56]. However, their complex dual-motif chemical structure is challenging to optimize, especially with respect to the structure of the linking group as subtle changes to structure can have considerable effects on biological function[22,57,58]. Furthermore, the process by which compounds are connected in FBDD, although theoretically guaranteed to yield superiorly active compounds, is rarely achieved despite decades-long efforts[20–22,30]. For these reasons we rationalize that studies of alternatively linked bivalent molecules, which exhibit a broad range of potency, would reveal structural insights that can be used to improve the processes in optimizing bivalent compounds. The bivalent ATP-allosteric EGFR inhibitors reported here provide a structural basis for linker potency and offer previously unconsidered strategies for linker design.

The molecules in this study are informative since they differ only in terms of linker structure and exhibit significantly different biochemical potencies (>10$^6$-fold in LRTM and LRTMCS, Fig. 4). The origins of this sizable potency range can be understood on the basis of how these compounds bind to the EGFR kinase domain. The "inward" allosteric benzo conformation of the *C*-linked scaffold **2** is found to best match the parent allosteric fragment **8** (Fig. 5h) and MD simulations indicate that stronger

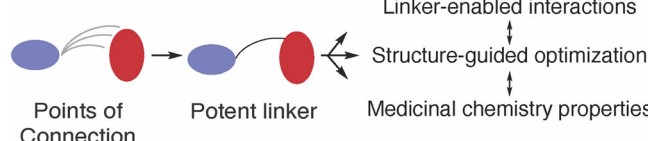

**Fig. 8 Points of connection serve as an effective strategy for linker optimization in early-phase drug discovery efforts.** To best optimize for compound binding, flexibility and fragment binding mode can be altered through alternative points of connection between the linker and fragment. Subsequent optimization of linker-enabled interactions as well as structure-guided optimization for potency and medicinal chemistry properties would follow using more conventional lead optimization strategies.

binding of **2** is due, in part, to structural complementarity with the allosteric site. Matching the binding modes of parent fragments is a well-appreciated objective in FBDD linking[28], which is consistent with the structures and biochemical potencies of **1** versus **2**. Interestingly, the binding mode "inward" conformation of **2** most closely resembles the allosteric-only **8** despite possessing the *N*-linked structure of **1**, and demonstrates how minute alterations in linker structure can manifest as major conformational differences in target binding sites[50]. We further characterize the linker structure to be the key factor in enabling benzo mobility provided by variations in crystallographic B-factors of **1** and **2** in cocrystal structures (Fig. 6a–b, Supplementary Fig. S8), MM-GBSA ligand energy calculations (Supplementary Table S3, Supplementary Fig. S10) and rotatable bond torsion profiles (Supplementary Fig. S12), which attributes the enhanced binding of **2** to a single unique C-C bond within the linker (orange in Fig. 6g, Supplementary Fig. S12). This potency-enabling aspect of the structure of **2** is only possible by shifting the linker connection (*N*- to *C*-linked) on the benzo motif (Table 1, Figs. 4 & 7). Several previous studies have highlighted the importance of enabling proper flexibility for effective linkers[30,59], but to our knowledge our compounds are distinctive with respect to the range of potency (>$10^6$-fold) and binding energy despite the relatively subtle differences in linker structure (Fig. 4, $\Delta G$ Supplementary Table S3).

Given the notable potency enhancement between the *N*- and *C*-linked scaffolds, we further considered how these compounds may inform strategies for more efficient linker optimization. Several recent reviews on FBDD linking offer perspectives from dozens of examples, but very little is understood regarding efficient design strategies beyond iterative trial and error[20–22,30,60]. If we consider the evolution of the *N*- to *C*-linked scaffold, the structural basis for the large potency enhancement is related to how the linker is connected with the benzo allosteric motif. The shift in the linker "point of connection" from **1** to **2** introduces several attributes that are not necessarily possible in varying what is considered to be more conventional linker structural properties, such as length or alternative functional groups. To our knowledge, linker optimization of this sort has not been discussed previously, however, it is worth noting the parallels with pioneering work by Fesik and co-workers and their work pertaining to Bxl-x$_L$ inhibitors[61,62]. Specifically their linked lead compound, connected through a central *trans*-olefin, was shown to exhibit improved binding by a shift to a linear acylsulfonamide linker structure that varies with respect to different point of connection, and ultimately lead to the FDA-approved drug venetoclax[62,63]. In terms of general optimization strategies, we propose that once linked, early-phase exploration of points of connection to the linked motifs are likely to result in large changes to drug potency (Fig. 8). While understandably cumbersome in terms of synthesis, we rationalize that arriving at the optimal linker structure at the onset of a project is the most efficient scenario and as such would streamline lead optimization. Additionally, this strategy may be especially helpful in cases where structural information regarding the linked molecule and target is challenging to obtain.

The iterative nature of drug optimization makes developing medicines reliant on brute force structural explorations and serendipity[20,60,64], but results from our work offer strategies to more effectively optimize highly sensitive regions of bivalent inhibitors. The dissection of the structural basis of our differently linked molecules inform an alternative perspective to the best of our knowledge on linker design, and also demonstrates that insights can be afforded through studies of linker-dependent effects on three-dimensional binding and how they translate to function. We rationalize that multi-site model systems, such as EGFR and others, represent constructive frameworks for evaluating drug design strategies amenable to structure determination with the ultimate goal to streamline structure-based drug optimization. Importantly, the unique structural attributes of any model system must be taken into consideration when proposing general design strategies, which will be refined in future studies in diverse receptor model systems. Additionally, this work presents distinct examples of bivalent scaffolds unique within the diverse repertoire of EGFR tyrosine kinase inhibitors, which we show here are active in human NSCLC cell lines across the most prevalent oncogenic activating mutations (L858R, T790M, and exon19del). Future work will be directed toward expanding evaluations of bivalent inhibitor applicability broadly across the human kinome and improve our understanding of structure-based drug design.

## Methods

**Protein expression and purification.** The EGFR kinase domain (residues 696–1022) was cloned into pTriEx with an N-terminal 6xHis-glutathione S-transferase (GST) fusion tag followed by a TEV protease cleavage site. EGFR WT, L858R, L858R/T790M, L858R/T790M/C797S was expressed after baculoviral infection in SF9 cells and EGFR(T790M/V948R) was expressed in SF21 cells. Briefly, cells were pelleted and resuspended in lysis buffer composed of 50 mM Tris pH 8.0, 500 mM NaCl, 1 mM tris(2-carboxyethyl) phosphine (TCEP), and 5% glycerol. Cells were lysed via sonication prior to ultracentrifugation at >200,000 g for 1 h. Imidazole pH 8.0 was added to the supernatant for a final concentration of 40 mM and flowed through a column containing Ni-NTA agarose beads. The resin was washed with lysis buffer supplemented with 40 mM imidazole and eluted with lysis buffer containing 200 mM imidazole. Eluted EGFR kinase domain was dialyzed overnight in the presence of 5% (w/w) TEV protease against dialysis buffer containing 50 mM Tris pH 8.0, 500 mM NaCl, 1 mM TCEP, and 5% glycerol. The cleaved protein was passed through Ni-NTA resin to remove the 6xHis-GST fusion protein and TEV prior to size exclusion chromatography on a prep-grade Superdex S200 column in 50 mM Tris pH 8.0, 500 mM NaCl, 1 mM TCEP, and 5% glycerol. Fractions containing EGFR kinase of ≥95% purity as assessed by Coomassie-stained SDS-PAGE were concentrated to approximately 4 mg/mL as determined by Bradford assay or absorbance.

**Crystallization and structure determination.** EGFR(T790M/V948R) pre-incubated with 1 mM AMP-PNP and 10 mM MgCl$_2$ on ice was prepared by hanging-drop vapor diffusion over a reservoir solution containing 0.1 M Bis-Tris (pH = 5.5), 25% PEG-3350, and 5 mM TCEP (buffer A for crystals soaked with compound **1**) or 0.1 M Bis-Tris (pH = 5.7), 30% PEG-3350 TCEP (buffer B for crystals soaked with compound **2**). Drops containing crystals in buffer A and B were exchanged with solutions of both buffers containing ~1.0 mM **1** or **2** were exchanged three times

for an hour and then left to soak overnight. Crystals were flash frozen after rapid immersion in a cryoprotectant solution with buffer A or B containing 25% ethylene glycol. X-ray diffraction data on T790M/V948R-compound **1** crystals were collected at 100 K at the National Synchrotron Light Source II 17-ID-2 (wavelength 0.97933 Å)[65]. X-ray diffraction data of T790M/V948R-compound **2** crystals was collected at 100 K at the Advanced Light Source a part of the Northeastern Collaborative Access Team (NE-CAT) on Beamline 24-ID-C (wavelength 0.97918 Å). In both cases, diffraction data was processed and merged in Xia2 using aimless and dials. The structure was determined by molecular replacement with the program PHASER using the inactive kinase domain EGFR(T790M/V948R) kinase from our previous work excluding the LN3844 ligand (PDB 6WXN). Repeated rounds of manual refitting and crystallographic refinement were performed using WinCoot (v.0.9.6.EL), and Phenix (v.1.20.1-4487),. The inhibitor was modeled into the closely fitting positive $F_o - F_c$ electron density and then included in following refinement cycles. Statistics for diffraction data processing and structure refinement are shown in Table 3. The Ramachandran statistics are; Ramachandran favored 96.31% (**1**, PDB ID 8FV3) and 95.88 (**2**, PDB ID 8FV4), Ramachandran allowed 3.00% (**1**, PDB ID 8FV3) and 3.34% (**2**, PDB ID 8FV4), Ramachandran outliers 0.69% (**1**, PDB ID 8FV3) and 0.77% (**2**, PDB ID 8FV4). Due to a mixture of difference map density with contributions from both AMP-PNP and **2** in Chain C (PDB ID 8FV4), we elected to leave this chain without bound ligands.

**Time-dependent Kinase Inhibition Assays**. Biochemical assays were performed with commercially available EGFR WT, cytoplasmic domain (669–1210), GST-tagged, Carna (Cat#/Lot#: 08-115/21CBS-0127H), EGFR [L858R], cytoplasmic domain (668-end), GST-tagged, SignalChem (Cat#/Lot#: E10-122BG/D2411-4), EGFR [T790M/L858R] cytoplasmic domain (669–1210), GST-tagged, Carna (Cat#/Lot#: 08-510/12CBS-0765M). Reactions were performed with kinase domain enzyme concentrations of WT EGFR at 2.0 nM, LR at 1.0 nM, and LRTM at 2.0 nM in final solutions of 52 mM HEPES pH 7.5, 1 mM ATP, 0.5 mM TCEP, 0.011% Brij-35, 0.25% glycerol, 0.1 mg/ml BSA, 0.52 mM EGTA, 10 mM $MgCl_2$, 15 μM Sox-based substrate (AQT0734). BSA was not included in this experiment to prevent interference with irreversible inhibitor characterization via off-target binding. All reactions were run for 240 min at 30 °C. Time-dependent fluorescence from the Sox-based substrate was monitored in PerkinElmer ProxiPlate-384 Plus, white shallow well microplates (Cat. #6008280) Biotek Synergy Neo 2 microplate reader with excitation (360 nm) and emission (485 nm) wavelengths. **4** was dosed between 0 and 10 μM in 24-point curves with 1.5-fold dilutions. Fluorescence, determined with identical reactions but lacking purified enzyme or crude cell lysate was subtracted from the total fluorescence signal for each time point, with both determined in duplicate, to obtain corrected relative fluorescence units (RFU). Corrected RFU values then were plotted vs. time and the reaction velocity for the first ~40 min (initial reaction rates) was determined from the slope using GraphPad Prism (La Jolla, CA) with units of RFU/min.

**HTRF Assays**. Biochemical assays for EGFR domains were carried out using a homogeneous time-resolved fluorescence (HTRF) KinEASE-TK (Cisbio) assay, as described previously[66]. Assays were optimized for ATP concentration of 100 μM with enzyme concentrations WT EGFR 10 nM, L858R 0.1 nM, L858R/T790M at 0.02 nM and L858R/T790M/C797S at 0.02 nM. Inhibitor compounds in DMSO were dispensed directly in 384-well plates with the D300 digital dispenser (Hewlett Packard) followed

immediately by the addition of aqueous buffered solutions using the Multidrop Combi Reagent Dispenser (Thermo Fischer). Compound $IC_{50}$ values were determined by 11-point inhibition curves (from 1.0 to 0.00130 μM) in triplicate. The FRET signal ratio was measured at 665 and 620 nm using a PHERAstar microplate reader (BMG LABTECH). The data was graphically displayed using GraphPad Prism version 9.0, (GraphPad software). The curves were fitted using a non-linear regression model with a sigmoidal dose response.

**Cellular Antiproliferative Activity Assays**. H1975 and HCC827 cells were obtained from the lab of Dr. Pasi Jänne (Dana-Farber Cancer Institute, 2022) cultured at 37 °C in RPMI 1640 media (Corning, 1004-CV) supplemented with 10% fetal bovine serum (Tissue Culture Biologicals, 35-010-CV) and 1% penicillin and streptomycin (P/S, Corning, 30-002-CI) and seeded overnight in a 96-well plate at a density of 60000 cells/mL. Cells were dosed with inhibitors to a final DMSO 0.5% in triplicate for 37 °C for 72 h. Cellular inhibition of growth was assessed by MTT viability assay according to the manufactures protocol (OZ Biosciences). Parental Ba/F3 cells was a generous gift from the laboratory of Dr. David Weinstock (in 2014), Dr. Pasi Jänne (2020) both of the Dana-Farber Cancer Institute and was used to generate the wildtype EGFR, L858R, and L858R/T790M EGFR mutant Ba/F3 cells[52,67]. Ba/F3 cells were all cultured in RPMI1640 media with 10% fetal bovine serum and 1% penicillin and streptomycin. The Ba/F3 cell lines were tested negative for *Mycoplasma* using Mycoplasma Plus PCR Primer Set (Agilent) and were passaged and/or used for no longer than 4 weeks for all experiments. Assay reagents were purchased from MilliporeSigma (Cat# R7017-5G). Ba/F3 cells were plated and treated with increasing concentrations of inhibitors in triplicate for 72 h. Compounds were dispensed using the Tecan D300e Digital Dispenser. Cellular growth or the inhibition of growth was assessed by resazurin viability assay to a final 1% DMSO. All experiments were repeated at least 3 times and values were reported as an average with standard deviation.

**Western blotting**. H1975, H3255, H3255GR and HCC827 lung adenocarcinoma cells (obtained from Dr. Pasi Jänne in Dana-Farber Cancer Institute, 2022) were cultured in RPMI 1640 media (Corning, 1004-CV) supplemented with 10% FBS (Tissue Culture Biologicals, 35-010-CV) and 1% penicillin-streptomycin (P/S, Corning, 30-002-CI). All cell lines were tested negative for *Mycoplasma* using Myco-Sniff-Rapid™ Mycoplasma Luciferase Detection Kit Plus PCR Primer Set (MP biomedicals, 0930504-CF) and were passaged and/or used for no longer than 4 weeks for all experiments H1975 and HCC-827 cells were seeded in 6-well plates with 400,000 cells per well and incubated overnight. H3255 and H3255GR cells were seeded in 12-well plates with 200,000 cells per well and were grown to confluency after 48 h. Cells were treated the next day after replacing fresh media for 6 h as indicated in the figure legends. Culture medium was removed, cells washed with PBS, and lysed with lysis buffer containing 5 M NaCl, 1 M TRIS pH 8.0, 10% SDS, 10% Triton X-100 and a tablet of protease and phosphatase inhibitor. Protein lysate concentration was analyzed using Pierce BCA kit (ThermoFisher, 23225). Protein samples (10 μg) were resolved on hand-cast Criterion Tris-HCl protein gels. Primary antibodies used; phospho-EGFR (Tyr1068; #2234, 1:1,000), EGFR (#4267; 1:1,000), phospho-AKT (Ser473; #4060, 1:1,000), AKT (#9272, 1:1,000), phospho-ERK1/2 (Thr202/Tyr204; #4370, 1:1,000), and ERK1/2 (#4695, 1:1,000) antibodies; were purchased from Cell Signaling Technology. Secondary Goat anti-rabbit IgG starbright blue 700 (Biorad, 64484700) and Anti-tubulin hFAB Rhodamine Tubulin (Bio-Rad,

64512248). Western blots were visualized with the ChemiDoc MP imager (Bio-Rad) utilizing the Image Lab Touch Software (version 2.4.0.03). All Western blot images are representatives of at least 3 independent replicate experiments.

**Metabolic stability in Human Liver Microsomes (HLM).** Pooled liver microsomes from humans (male) were purchased from *Sekisui XenoTech, LLC*, Kansas City, KS, USA. Metabolic stability assays were performed in the presence of an NADPH-regenerating system consisting of 5 mM glucose-6-phosphate, 5 U/mL glucose-6-phosphate dehydrogenase, and 1 mM $NADP^+$. Liver microsomes (20 mg/mL), NADPH-regenerating system, and 4 mM $MgCl_2 \cdot 6 H_2O$ in 0.1 M TRIS-HCl-buffer (pH 7.4) were preincubated for 5 min at 37 °C and 750 rpm on a shaker. The reaction was started by adding the preheated compound at 10 mM resulting in a final concentration of 0.1 mM. The reaction was quenched at selected time points (0, 10, 20, 30, 60, and 120 min) by pipetting 100 μL of internal standard (ketoprofen) at a concentration of 150 μM in acetonitrile. The samples were vortexed for 30 s and centrifuged (21910 relative centrifugal force, 4 °C, 20 min). The supernatant was used directly for LC-MS analysis. All compound incubations were conducted at least in triplicates. Additionally, a negative control containing BSA (20 mg/mL) instead of liver microsomes and a positive control using verapamil instead of compound were performed. A limit of 1% organic solvent during incubation was not exceeded. Sample separation and detection were performed on an *Alliance 2695 Separations Module* HPLC system (*Waters Corporation*, Milford, MA, USA) equipped with a *Phenomenex Kinetex* 2.6 μm XB-C18 100 Å 50 ×3 mm column (*Phenomenex Inc*., Torrance, CA, USA) coupled to an *Alliance 2996 Photodiode Array Detector* and a *MICROMASS QUATTRO micro API* mass spectrometer (both *Waters Corporation*, Milford, MA, USA) using electrospray ionization in positive mode. Mobile phase A: 90% water, 10% acetonitrile and additionally 0.1% formic acid (v/v), mobile phase B: 100% acetonitrile with additionally 0.1% formic acid (v/v). The gradient was set to: 0–2.5 min 0% B, 2.5–10 min from 0 to 40% B, 10–12 min 40% B, 12.01–15 min from 40 to 0% B at a flow rate of 0.7 mL/min. Samples were maintained at 10 °C, the column temperature was set to 20 °C with an injection volume of 5 μL. Spray, cone, extractor, and RF lens voltages were at 4 kV, 30 V, 8 V and 2 V, respectively. The source and desolvation temperatures were set to 120 °C and 350 °C, respectively, and the desolvation gas flow was set to 750 L/h. Data analysis was conducted using *MassLynx 4.1* software (*Waters Corporation*, Milford, MA, USA).

**Compound Docking.** Computer-aided compound docking was performed with GLIDE (GlideScore version SP5.0 Schrödinger, LLC, New York, NY, 2021, re. 2021-2) with standard precision, the Maestro 12.8.117 portal. The receptor grid was generated from the EGFR(T790M/V948R) kinase domain from Chain D of PDB ID 8FV4 (compound **2**) with the omitted ligand using the Protein Preparation Wizard[68]. Compound **4** was prepared with LigPrep (OPLS4 force field, Epik pH = 7.0 ± 2). The best binding poses were ranked on the basis of the lowest docking and glide-score values[69,70].

**Modeling and structure preparation for MD simulations.** Molecular modeling was conducted using Maestro (Schrödinger Release 2023-1, Schrödinger LLC, New York, NY, 2021) and the OPLS4 force field[71]. The crystal structures **1** (PDB ID 8FV3) and **2** (PDB ID 8FV4) were utilized for modeling the complexes of compound **1** and compound **2**, respectively. Prior to modeling, the protein structures were prepared using the Protein

Preparation Wizard[68] (Maestro 2021.4, Schrödinger LLC, New York, NY, USA) with default settings, which involved adding hydrogen atoms and correcting any missing side chains. In the compound **1** complex, the disordered residues in the activation loop required were rebuilt as following: the A859-A871 region was rebuilt based on the chain B of the 8FV4 crystal structure using the chimera homology modeling approach, followed by further minimization of the region using the OPLS4 force field within a selected interval and a 3 Å region around the selected residue interval. Additionally, the E872-E874 residues were added using Maestro's cross-link proteins tool, followed by region minimization in the OPLS4 force field. For the compound 2 complex, the L862-A871 region of the activation loop was rebuilt based on the chain B of the 8FV4 crystal structure using chimera homology modeling. Similarly, the E872-K875 residues and S784 residue were added using Maestro's cross-link proteins tool, followed by region minimization in the OPLS4 force field within a selected interval and a 3 Å region around the selected residue interval. The initial validation of the individual models was assessed using the Ramachandran plot.

**Molecular Dynamics Simulations.** Desmond MD engine[72] was used for the MD simulations with OPLS4[71] force field. The system was solvated in an orthorhombic box (minimum distance of 10 Å to the edges from the protein). A temperature of 300 K was used for membrane patch pre-equilibration. The water was described with the TIP3P[73] model. The final systems comprised ~44 k atoms. All simulations were run in the NpT ensemble (T = 310 K, Nosé-Hoover method; p = 1.01325 bar, Martyna-Tobias-Klein method) with default Desmond settings. Reversible reference system propagator algorithms (RESPA) integrator with 2 fs, 2 fs, and 6 fs timesteps were used for bonded, near and far, respectively. Short-range coulombic interactions were calculated using 1 fs time steps and 9.0 Å cut-off value, whereas long-range coulombic interactions were estimated using the Smooth Particle Mesh Ewald method, which is a sufficiently good approximation to treat long-range interactions on large timescales[74]. The system was relaxed using the default Desmond protocol before the production simulation.

A total of 20 simulation replicas were prepared individually, each simulated with a random seed for a duration of 1μs for compound **1** and compound **2**. This resulted in a cumulative simulation time of 20 μs (10 replicas x 1μs x 2 compounds). All production simulations were conducted using consistent settings. The simulation interaction diagram provided by the Maestro package (Schrödinger, LLC, New York, NY) served as the foundation for the analysis of the simulations.

**Interaction analyses.** Protein–ligand interactions, as well as hydrophobic interaction frequency, RMSD, and torsional conformations of rotatable bonds were analyzed by the Simulation Interaction Analysis tool (scripts: event_analysis.py; analyze_simulation.py) (Schrödinger LLC). The default settings were used in the definition of the interactions, where the following parameters were applied: H-bonds, a distance of 2.5 Å between the donor and acceptor with ≥120 and ≥90° for donor and acceptor angles, respectively; π–cation interactions, a 4.5 Å distance between the positively charged and aromatic group; π–π interactions, stacking of two aromatic groups face-to-face or face-to-edge; water bridges, a distance of 2.8 Å between the donor and acceptor with ≥110 and ≥90° for donor and acceptor angles, respectively.

**MM-GBSA.** The molecular mechanics energies with generalized Born and surface area continuum solvation (MM-GBSA) were

calculated with Prime Thermal MM/GBSA[75,76]. Each 2nd frame of MD was used for MM–GBSA calculations (5010 complexes proceeded for an individual complex x 2 ligands). MM-GBSA calculations report energies for the ligand, receptor, and complex structures as well as energy differences relating to strain and binding and are broken down into contributions from various terms in the energy expression.

**WaterMap**. WaterMap[77,78] simulations were using Maestro, and the system was solvated in TIP3P water box extending at least 10 Å beyond the truncated protein in all directions. 5 ns MD simulation was performed, following a standard relaxation protocol, and the water molecule trajectories were then clustered into distinct hydration sites. Entropy and enthalpy values for each hydration site were calculated using inhomogeneous solvation theory.

**Conformational analysis**. Conformational analysis of compound 1 and 2 was assessed with Conformational Search tool from MacroModel module (Schrödinger LLC). Analysis was conducted with OPLS4[71] force field force field in a water solvent was employed for the analysis. A mixed torsional/Low mode sampling approach was chosen with the default settings. The Polak-Ribier Conjugate Gradient[79] (PRCG) method with restarts every 3 N iterations (maximum of 2500 iterations) was utilized for energy minimization, with a convergence threshold of 0.05. For compound 1, a total of 139 conformers were generated, while for compound 2, 278 conformers were generated. The ligand conformation that best matched the crystal structure was determined by SMARTS superimposition of the structure scaffolds with the reference ligand conformation. The selection of the final conformation was justified based on the lowest superimposition root mean square deviation (RMSD) values obtained from both conformational datasets.

**Data Visualization**. Results were plotted with Seaborn library for Python 3.8.8[80]. Protein structures were visualized with PyMOL (The PyMOL Molecular Graphics System, Version 2.5.2 Schrödinger, LLC.) Graphical representations of figures were arranged using Adobe Illustrator©.

**Chemistry**. All starting materials, reagents, and (anhydrous) solvents were commercially available and were used as received without any further purification or drying procedures unless otherwise noted. All NMR spectra were obtained with Bruker Avance 200 MHz and Bruker Avance 400 MHz spectrometers or with a Bruker Avance 600 MHz spectrometer (NMR Department, Institute of Organic Chemistry, Eberhard-Karls-Universität Tübingen) or Bruker Ascend 400 MHz and Bruker Ascend 500 MHz (Magnetic Resonance Center, Department of Chemistry, SUNY at Buffalo). Solvents for NMR are noted in the experimental procedures for each compound. Residual solvent peaks were used to calibrate the chemical shifts. Chemical shifts (δ) are reported in parts per million. Mass spectra were obtained by Advion TLC-MS (ESI) and from the MASS Spectrometry Department (ESI-HRMS), Institute of Organic Chemistry, Eberhard-Karls-Universität Tübingen or by Thermo Scientific LTQ XL Linear Ion Trap Mass Spectrometer (Small Instrument Center, Department of Chemistry, SUNY at Buffalo). The purity of the tested compounds was determined via HPLC analysis on an Agilent 1100 Series LC with a Phenomenex Luna C8 column (150 × 4.6 mm, 5 µm), and detection was performed with a UV diode array detector (DAD) at 254 and 230 nm wavelengths via elution condition (A) or on an Agilent 1200 series LC with an Agilent Eclipse XDB-C18 column (4.6×150 mm, 5 µm) and detection was performed using an Agilent 1200 series Multiple

Wavelength Detector (MWD) at 254, 280, and 305 nm wavelengths via elution condition (B) and was >95%. Elution was carried out with the following conditions: (A) 0.01 M $KH_2PO_4$, pH 2.30 (solvent A), and MeOH (solvent B), 40% B to 85% B in 8 min, 85% B for 5 min, 85% to 40% B in 1 min, 40% B for 2 min, stop time of 16 min, 5 µL injection volume, flow rate of 1.5 mL/min, and 25 °C oven temperature or (B) 60% MeOH, 40% 0.1% formic acid in $H_2O$, stop time of 16 min, 5 µL injection volume, flow rate of 1.5 mL/min, 25 °C oven temperature. Thin-layer chromatography (TLC) analyses were performed on fluorescent silica gel 60 F254 plates (Merck) and visualized via UV illumination at 254 and 366 nm. Column chromatography was performed on Davisil LC60A 20–45 µm silica from Grace Davison as the stationary phase and Geduran Si60 63–200 µm silica from Merck for the precolumn using an Interchim PuriFlash XS 420 automated flash chromatography system.

**Reporting summary**. Further information on research design is available in the Nature Portfolio Reporting Summary linked to this article.

## Data availability

All data are available from the authors upon reasonable request. All crystal structures are publicly available from the Protein Data Bank via the accession codes 8FV3 (Compound 1) and 8FV4 (compound 2). The datasets generated during the in silico work are available in the Zenodo repository (DOI 10.5281/zenodo.8020238). The available data includes the molecular dynamic trajectories for 1 and 2, along with the raw data on interaction patterns, separated for interaction components (i.e. Hydrophobic interactions/Hydrogen Bonding), full-component MM/GBSA tables, WaterMap Pymol sessions, and raw output of ligand conformational analysis. All other data are available from the corresponding authors (or other sources, as applicable) on reasonable request. Supplementary Data 1 contains copies of the $^1$H and $^{13}$C spectra of isolated new compounds, and the raw data are available upon request. Supplementary Data 2 contains percent activity data featured in Fig. 4. Supplementary Data 3 and 4 contain atomic coordinates and other crystallographic information for the cocrystal structures of 1 (PDB ID 8FV3) and 2 (PDB ID 8FV4) in complex with EGFR(T790M/V948R), respectively.

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

## Acknowledgements

We acknowledge support from startup funds from The State University of New York (D.E.H.) and support by the National Center for Advancing Translational Sciences of the National Institutes of Health under award Number UL1TR001412-08 (BTC K Scholar Award to D.E.H.). S.A.L. and iFIT are funded by the Deutsche Forschungsgemeinschaft (DFG, German Research Foundation) under Germany's Excellence Strategy (EXC 2180-390900677). TüCAD2 is funded by the Federal Ministry of Education and Research (BMBF) and the Baden-Württemberg Ministry of Science as part of the Excellence Strategy of the German Federal and State Governments. The authors acknowledge CSC-IT Center for Science, Finland for computational resources. National Institutes of Health Grants R01CA201049, R01CA116020, and R35CA242461 (to M.J.E), Roswell Park Alliance Foundation (P.A.H.). The content is solely the responsibility of the authors and does not necessarily represent the official views of the NIH. T.S.B is supported by a Ruth L. Kirschstein National Research Service Award (5F32CA247198-02). This work was partially done in the Drug Discovery Core Facility of Roswell Park Comprehensive Cancer Center supported by National Cancer Institute (R01CA197967 to K.V. Gurova and P30CA016056 to Roswell Park Cancer Center). This work is based on research utilizing resources of the Frontier Microfocusing Macromolecular Crystallography (17-ID-2, FMX) beamline at National Synchrotron Light Source II at Brookhaven National Laboratory to Block Allocation Group 308246. This work is based upon research conducted at the Northeastern Collaborative Access Team beamlines (P30 GM124165, P41 GM103403) utilizing resources of the Advanced Photon Source at the Argonne National Laboratory (DE-AC02-06CH11357). We thank Mr. Bryan Renzoni for contributions to synthesis. We thank Dr. David Hangauer, Dr. Andrew Gulick, Dr. Steven Diver, Dr. Thales Kronenberger and Frederik Hacker for insightful comments and discussions. We are grateful to Dr. Michael Malkowski and Dr. Liang Dong for SF21 cells and access to their tissue culture lab. We also acknowledge Dr. Diana Monteiro and Dr. Edward Snell for access to laboratory space and equipment for protein purification and crystallization resources. We acknowledge support from the Open Access Publication Fund of the University of Tübingen.

## Author contributions

F.W., B.C.O., E.S., T.D., C.D.P., I.K.S., B.T.O, S.P.C., T.S.B., A.R., B.B., D.U., T.S., E.M., E.S., M.J.E., P.A.H., A.P., S.A.L., D.E.H. conceived and designed the experiments; A.P. and E.S. conducted MD simulations with energy, torsional, and water profiling; F.W. and B.T.O. performed synthesis; B.C.O., T.D., C.D.P., S.P.C., I.K.S., T.S.B., T.S. and B.B. performed biological assays; D.U. and E.M. determined time-dependent kinetic parameters; B.C.O., C.D.P. T.S.B. and D.E.H. performed X-ray crystallography; S.P.C. performed molecular docking; A.R. performed metabolic stability experiments; F.W., B.C.O., S.A.L. and D.E.H. interpreted the data; F.W., B.C.O., S.A.L. and D.E.H. wrote the paper. The manuscript was written through the contributions of all authors. All authors have given approval to the final version of the manuscript.

## Funding

## Competing interests

D.E.H., F.W., B.C.O, C.D.P., A.R., and S.A.L. are inventors on a provisional patent (63/483,871 filed 2/8/2023) comprising aspects of the molecules in this study. D.E.H. has been a consultant to Addition L.P. and is a member of the advisory board for Tropocan Therapeutics. D.A.U., E.M.S., and E.W.M. are employees and shareholders of Assay-Quant Technologies, Inc., a provider of life science tool products and services that support drug discovery, including the PhosphoSens platform used to generate the data presented here. M.J.E. is a consultant to Novartis, and the Eck laboratory receives or has received research funding from Novartis, Takeda, Sanofi, and Arbella. All other authors declare no competing interest.
