## [Peer Review File · Communications Chemistry]

Reviewers' comments:

Reviewer #1 (Remarks to the Author):

This manuscript describes the discovery of highly potent EGFR inhibitors by simultaneously targeting the ATP binding pocket and an adjacent allosteric site. One of the C-linked molecules, compound 4, exhibited significantly higher potency (~60 pM) against EGFR L858R/T790M and EGFR L858R/T790M/C797S resistant mutants, which is approximately 1000~100000 fold more potent than the original ATP or allosteric binders. The authors further utilized co-crystal structures and computational molecular simulation approaches to gain in-depth understanding about the contribution of the linker to the superadditive effect of the bivalent molecule. The results are very interesting and I support the publication of this manuscript. However, there are some issues the authors may consider:

- 1) The bivalent exhibited significantly stronger activity than most of the reference molecules in the manuscript, but its cellular effect is obviously lower. The authors may give some explanation on this inconsistency.
- 2) It is a general information that HCC827 cells harbor EGFR del19, which does not have an allosteric pocket as that in other EGFR mutants, how to explain the strong inhibition against the phosphorylation of EGFR in the cell-based assay?
- 3) The authors discussed that their results will provide a general approach for the linker selection in other bivalent inhibitor design, but this reviewer cannot agree on this point. It is generally accepted that the length and shape of the linker, the linking position will have a huge impact on the activity of the corresponding bivalent ligands. The linker selection highly depends on the geometry, space and electronic properties of the "channel" between the two distinct pockets in different proteins.

Reviewer #2 (Remarks to the Author):

In this manuscript, Wittlinger et al. present a series of EGFR inhibitors developed by integrating both ATP-competitive and allosteric inhibitor elements. These bivalent inhibitors exhibited robust biochemical activities against a range of mutant EGFRs. Notably, the efficiency of these inhibitors dramatically improved depending on the 'link-out' position. Co-crystal structures were also provided, offering explanations for the varying performance of these inhibitors. The manuscript is well-organized and a pleasure to read. However, while inhibitors 2 and 3 displayed high potency in biochemical assays, they achieved only submicromolar activities in cell growth inhibition tests. This discrepancy warrants further discussion. Given the lack of effective clinical therapeutic options for patients with C797S resistance, these inhibitors should be further evaluated in the C797S mutant Ba/F3 cells. This evaluation will help determine if these compounds are viable starting points for developing drugs to combat resistance to third-generation EGFR inhibitors.

Reviewer #3 (Remarks to the Author):

The manuscript offers a very interesting view on the strategies to be adopted in the design and

optimisation of bivalent EGFR kinase inhibitors. A topic of great interest in drug design especially for the well known target mutations found in some cancers.

I found this manuscript very good under different point of view. The workflow is well described and of great interest and the design of experiment has been carefully planned and developed. Discussion of results and conclusions are well presented and perfectly support the results showed. The use of triplicates and 11-point inhibition curve for HTRF, together with the other experimental choices, give the results a good statistical robustness.

I have just some minor suggestions that could improve the manuscript:

- In the Introduction it would be interesting to add some more information about pharmacological pros and cons of bivalent molecules compared to the use of two distinct molecules.

- In page 6 "the N-linked 1 was observed to be limitedly potent against WT and mutant". Which mutant do you refer to? can you please specify if all mutants or not?

- In page 8 "inactive (α C-helix "out") conformation (Figure 2B, , Supplementary Table S2)" please correct this typo (double comma).

- Page 9 What kind of docking did you adopt for compound 4? covalent docking? Please specify. Was docking used only for this compound?

- In the materials and methods section please give more details about docking in glide and ligand preparation parameters. How docking score and glide score were used? in consensus manner, were there some differences in values?

Reviewer #4 (Remarks to the Author):

Review of Manuscript COMMSCHEM-23-0465-T by Wittlinger et al.

The manuscript entitled „Linking ATP and allosteric sites to achieve superadditive binding with bivalent EGFR kinase inhibitors” by Wittlinger et al. reports a set of new bivalent epidermal growth factor receptor (EGFR) kinase inhibitors as well as their structural and functional characterization.

The authors describe the synthesis of four novel heterobifunctional compounds with different linker structures and characterize their potency (IC_{50}) on wild-type and mutant EGFR kinase domains using HTRF assays. Kinetic parameters for the irreversibly linked compound 4 are determined by a biochemical assay. Furthermore, two co-crystal structures of the EGFR kinase domain bound to compounds 1 or 2 are presented to characterize the protein-inhibitor interaction structurally in atomic detail and show the differences in binding between both compounds. In order to obtain a dynamic picture of the interactions, the authors conducted molecular dynamics (MD) simulations with both co-crystal structures and analyzed the interaction patterns, structural flexibility, binding energies, and

torsional profiles of the compounds. A model of compound 4 bound to the EGFR kinase domain was generated by protein-ligand docking, using one of the previously solved crystal structures. The effect of inhibitors 1-4 on the phosphorylation of kinase targets in different adenocarcinoma cell lines (*ex vivo*) was monitored by western blots of whole cell lysates and antiproliferative activities of compounds 2 and 4 were determined. Furthermore, the metabolic stability of compound 4 was shown in liver microsome assays. Finally, the data are discussed and an altered strategy for optimizing linkers of bivalent inhibitors is presented.

Overall, the presented results are novel and the manuscript will be important to scientists in the specific field of bivalent inhibitor development. The manuscript is well-written and clearly structured in most parts, and the conclusions drawn are supported by the data. However, the authors should invest more time to carefully prepare the main text and the figures containing structural data. The clarity, especially of the structural figures, should be improved with respect to coloring, labeling, and information content in the figure legends. I recommend publication of the manuscript after the authors have revised their manuscript to address the specific points below.

1. The sentence on p. 8, lines 244-246 contains an error. "The side chain of K745, the catalytic lysine, exhibits a "swing" toward the benzo ketone in the case of 2 binding opening a position on the imidazole, which now binds a solvent water (Figure 2D)." Obviously, a comma is missing between "binding" and "opening". Please correct.

2. The reference "Figure 1 C-D" at the end of the sentence on p. 11, lines 299-301 is incorrect. There is no Figure 1D as referenced because obviously Figure 2 C-D is meant. Please change accordingly.

3. To improve the clarity of the structural figures in general, the colors and labeling of the individual compounds/inhibitors should be consistent, at least per figure. Nearly all structural figures (X-ray, MD simulations, and docking) show both compounds in subfigures and are not consistently labeled. It is time-consuming to look up which compound is which in the figure legend. I would suggest that each subfigure containing structures is consistently labeled with the compound number in the color of the compound or in black throughout the manuscript and supplementary information. For example, in Figure 2c and d both compounds are colored in green and are labeled with black numbers. In Figure 2e, the coloring of compound 1 changed to yellow, while compound 2 is still green but now labeled with a green number. In Figure S2 compounds are not labeled at all and in the figure legend for Figure S2f obviously, the wrong compound number is noted (for details see point 5 below). Likewise, in Figure S7 and S8 compounds are not labeled. To solve the problem, I propose to reserve one color each for compounds 1 and 2 and use this coloring throughout the manuscript, but at least it should be consistent in one figure.

4. By calculating a simulated-annealing Fo-Fc omit map, the authors show unambiguously that compounds 1 or 2 are bound to the ATP- and allosteric site of the protein. However, it is necessary that the sigma levels of all electron density map snapshots are written in the legend of Figure S2. Furthermore, a consistent coloring of 2Fo-Fc and Fo-Fc electron density maps would improve the clarity of Figures S2B, C, E, and F. Currently, both 2Fo-Fc maps are colored in blue, while the Fo-Fc map for compound 1 is in purple and the one for compound 2 is again in blue, which is quite confusing. The

coloring should generally be consistent and explained in the figure legend, especially if different map types and structures with different compounds are shown side by side.

5. The figure legend of Figure S2f (p. S4, line 88) contains an error. It states that the Fo-Fc map of compound 1 is shown, however, it actually shows compound 2 with its omit map.

6. The legend of Figure S2 (p. S4, lines 83-88) does not contain a figure title, like all the other figures. Obviously, the title of the legend was accidentally shifted. I assume that the sentence "Electron density maps for cocrystal structures of 1 and 2 in complex with EGFR(T790M/V948R).", which is misplaced in the middle of the legend, actually is the title of the figure and should be moved to the beginning.

7. The crystallographic work is generally technically sound but the crystallographic table has to be improved. A lot of physical units in Table S2 reporting on the crystallographic data statistics are missing. Please add missing units for the wavelength [Å], the resolution range [Å], the unit cell dimensions [Å] and angles [°], the Wilson B-factor [Å²], the RMSD bonds [Å] and angles [°] as well as the average B-factors [Å²].

Furthermore, the units for the B-factor [Å²] are missing in Figure 3a and b, Supplementary Figure S8a and b as well as in their figure legends (p. 13, lines 332-333 and p. S11, lines 156-157).

8. The legend of Figure S3 (p. S5, lines 90-92) is incomplete. Please mention the colors of the compounds and the color of the activation loop in the figure legend.

9. Tables S3 and S4 in supplementary information contain a mixture of different font types. Since supplementary information is usually not copy-edited please unify font types.

10: Models derived from crystal structures are relaxed or equilibrated for some nanoseconds prior to the individual MD simulations, which is also mentioned in the methods section. For how long have the models been relaxed?

11. The RMSDs especially for the protein in the MD simulations (Figure S6) sometimes "suddenly jump" by > 2 Å (e.g. replica 2, 3, and 6 for compound 1 as well as replica 2 and 5 for compound 2). Could the authors explain what the reasons are for these large jumps when they check the models at these time points?

12. Figure S7 shows the result of docking compound 4 to the EGFR kinase domain. What are the yellow dashed lines that reach to the amino acid side chains or the compounds on one end but go nowhere on the other end?

One note on the information content of figures and supplementary figures that the authors may consider: Supplementary figures sometimes show the same information as the main figures. For example, Figure 3a and b (B-factor coloring of the ligand) shows the same information as Figure S8a and b. The only difference is the presence of the protein ribbon in Figure S8, which is, however, not discussed, mentioned in the text, or important otherwise. Another example is Figure 2e showing

compounds 1 and 2 superposed, and Figure S4c with exactly the same image but with an added angle and distance. The angle and distance could simply be added to Figure 2e.

Revision #1: Point-by-point reply to reviewers and editorial requests

Reviewer #1 (Remarks to the Author):

Comment: This manuscript describes the discovery of highly potent EGFR inhibitors by simultaneously targeting the ATP binding pocket and a adjacent allosteric site. One of the C-linked molecule, compound 4, exhibited significantly higher potency (~60 pM) against EGFR L858R/T790M and EGFR L858R/T790M/C797S resistant mutants, which is approximate 1000~100000 fold more potent than the original ATP or allosteric binders. The authors further utilized co-crystal structures and computational molecular simulation approaches to get in-depth understanding about contribution the linker to the superadditive effect of the bivalent molecule. The results are very interesting and I support the publication of this manuscript. However, there are some issues the authors may consider:

Reply: We appreciate the enthusiasm and comments provided by the reviewer, which have improved the quality of our manuscript.

Comment: 1) The bivalent exhibited significantly stronger activity than most of the reference molecules in the manuscript, but its cellular effect is obviously lower. The authors may give some explanation on this inconsistency.

Reply: This is an important observation, which deserves mention in the manuscript. We admit that the large molecular weight and the several H-bond donors / acceptors of these bivalent scaffolds presents challenges with respect to membrane permeability. In fact, we observe similar effects in cell-based experiments with our earlier bivalent scaffolds (Wittlinger, et al., J Med Chem 2022 <https://doi.org/10.1021/acs.jmedchem.1c00848>), which were confirmed to be poorly membrane permeable on the basis of size and molecular weight. We have added a sentence on page 14 indicating this inconsistency and the need for structural optimization for improved membrane permeability.

Comment: 2) It is a general information that HCC827 cells harbor EGFR del19, which does not have an allosteric pocket as that in other EGFR mutants, how to explain the strong inhibition against the phosphorylation of EGFR in the cell-based assay?

Reply: We admit to initially being surprised by this observation given the lack of activity exhibited by allosteric inhibitors against this deletion mutation. At present, limited is known regarding the structural basis for how TKIs generally inhibit these prevalent deletion mutations, although recent studies have offered important new insights (e.g. <https://doi.org/10.1038/s41467-022-34398-z>, <https://doi.org/10.1073/pnas.2206588119>, <https://doi.org/10.7554/eLife.65824>), no experimental evidence exists at present that confirms that a bivalent inhibitor can bind this deletion mutation. MD simulations by Galadas et al., 2021 indicate that this delE746-A750 can adopt a thermodynamically accessible α C-helix out conformation that theoretically would be capable of binding bivalent ATP-allosteric inhibitors. While preparing this manuscript, we have confirmed that similar, but structurally distinct ATP-

allosteric bivalent EGFR inhibitors with indolinone allosteric pocket groups also target this deletion mutation in HCC827 cells ([10.26434/chemrxiv-2023-zzfc3](https://doi.org/10.26434/chemrxiv-2023-zzfc3)) adding further evidence that our observations here are robust and warrant additional structural investigation to confirm these binding modes and EGFR mutant kinase conformations. We have added a sentence and new citation to Galdadas et al on page 14 of the manuscript speculating as to the possibility of a putative inactive conformation of this highly prevalent EGFR activating mutation.

Comment: 3) The authors discussed that their results will provide a general approach for the linker selection in other bivalent inhibitor design, but this reviewer can not agree on this point. It is generally accepted that the length and shape of linker, the linking position will have huge impact on the activity of the corresponding bivalent ligands. The linker selection highly depends on the geometry, space and electronic properties of the “channel” between the two distinct pockets in different proteins.

Reply: We generally agree with the reviewer that the precise selection of the linker group for any fragment-based linker scenario is likely not to resemble the particularities of the EGFR kinase domain. However, we believe the most applicable insights from these studies relevant to linking generally has to do with the nature of the linking optimization strategy. As we present in the discussion (pages 15-17), the point of connection by which the fragments are linked would be ideal starting points for obtaining the maximally active compound, which would be subsequently tuned through linker length, shape, and composition optimization. We have added an extra statement on page 17 that highlights how unique structural attributes of any model systems must be considered when proposing general design strategies and motivate future experiments for refined drug design protocols.

Reviewer #2 (Remarks to the Author):

Comment: In this manuscript, Wittlinger et al. present a series of EGFR inhibitors developed by integrating both ATP-competitive and allosteric inhibitor elements. These bivalent inhibitors exhibited robust biochemical activities against a range of mutant EGFRs. Notably, the efficiency of these inhibitors dramatically improved depending on the 'link-out' position. Cocrystal structures were also provided, offering explanations for the varying performance of these inhibitors. The manuscript is well-organized and a pleasure to read.

Reply: We are grateful for the reviewer’s positive reception of our manuscript and appreciate the opportunity to address the point below toward a further improved study.

Comment: 1) However, while inhibitors 2 and 3 displayed high potency in biochemical assays, they achieved only submicromolar activities in cell growth inhibition tests. This discrepancy warrants further discussion. Given the lack of effective clinical therapeutic options for patients with C797S resistance, these inhibitors should be further evaluated in the C797S mutant Ba/F3 cells. This evaluation will help determine if these compounds are viable starting points for developing drugs to combat resistance to third-generation

EGFR inhibitors.

Reply: We have included L858R/T790M/C797S Ba/F3 antiproliferative data for **2** and **4** in an expanded Supplemental Table S7. In reply to reviewer 2, we have added a comment about the inconsistency regarding the cellular activity in these antiproliferative experiments (see comment above). Coupled to this statement, we have included a comment regarding the activity of our bivalent inhibitors against C797S mutant Ba/F3 cell on page 14.

Reviewer #3 (Remarks to the Author):

Comment: The manuscript offer a very interesting view on the strategies to be adopted in the design and optimisation of bivalent EGFR kinase inhibitors. A topic of great interest in drug design especially for the well known target mutations found in some cancers. I found this manuscript very good under different point of view. The workflow is well described and of great interest and the design of experiment has been carefully planned and developped. Discussion of results and conclusions are well presented and perfectly support the results showed. The use of triplicates and 11-point inhibition curve for HTRF, together with the other experimental choices, give the results a good statistical robustness.

Reply: We thank the reviewer for their positivity of our work and points below that have led to an overall improvement to the manuscript.

Comment: I have just some minor suggestions that could improve the manuscript: 1) In the Introduction it would be interesting to add some more information about pharmacological pros and cons of bivalent molecules compared to the use of two distinct molecules.

Reply: We have added a sentence to page 3 of the introduction discussing pros and cons of bivalent molecules.

Comment: - In page 6 "the N-linked 1 was observed to be limitedly potent against WT and mutant". Which mutant do you refer to? can you please specify if all mutants or not?

Reply: We have adjusted this statement to refer to "all tested EGFR mutants" on page 6.

Comment: - In page 8 "inactive (α C-helix "out") conformation (Figure 2B, , Supplementary Table S2)" please correct this typo (double comma).

Reply: We have corrected this typo.

Comment: - Page 9 What kind of docking did you adopt for compound 4? covalent docking? Please specify. Was docking used only for this compound?

Reply: Docking was only done for this compound to confirm the binding mode of the covalent analogue **4**. We utilized the basic reversible poses enabled in GLIDE from Schrödinger.

Comment: - In the materials and methods section please give more details about docking in glide and ligand preparation parameters. How docking score and glide score were used? in consensus manner, were there some differences in values?

Reply: We have included docking and glide score for the most favorable docking pose of **4** (Figure S7). We have also added details pertaining to how we carried out LigPrep and GLIDE.

Reviewer #4 (Remarks to the Author):

Comment: The manuscript entitled „Linking ATP and allosteric sites to achieve superadditive binding with bivalent EGFR kinase inhibitors” by Wittlinger et al. reports a set of new bivalent epidermal growth factor receptor (EGFR) kinase inhibitors as well as their structural and functional characterization.

The authors describe the synthesis of four novel heterobifunctional compounds with different linker structures and characterize their potency (IC₅₀) on wild-type and mutant EGFR kinase domains using HTRF assays. Kinetic parameters for the irreversibly linked compound **4** are determined by a biochemical assay. Furthermore, two co-crystal structures of the EGFR kinase domain bound to compounds **1** or **2** are presented to characterize the protein-inhibitor interaction structurally in atomic detail and show the differences in binding between both compounds. In order to obtain a dynamic picture of the interactions, the authors conducted molecular dynamics (MD) simulations with both co-crystal structures and analyzed the interaction patterns, structural flexibility, binding energies, and torsional profiles of the compounds. A model of compound **4** bound to the EGFR kinase domain was generated by protein-ligand docking, using one of the previously solved crystal structures. The effect of inhibitors **1-4** on the phosphorylation of kinase targets in different adenocarcinoma cell lines (ex vivo) was monitored by western blots of whole cell lysates and antiproliferative activities of compounds **2** and **4** were determined. Furthermore, the metabolic stability of compound **4** was shown in liver microsome assays. Finally, the data are discussed and an altered strategy for optimizing linkers of bivalent inhibitors is presented.

Overall, the presented results are novel and the manuscript will be important to scientists in the specific field of bivalent inhibitor development. The manuscript is well-written and clearly structured in most parts, and the conclusions drawn are supported by the data. However, the authors should invest more time to carefully prepare the main text and the figures containing structural data. The clarity, especially of the structural figures, should be improved with respect to coloring, labeling, and information content in the figure legends. I recommend publication of the manuscript after the authors have revised their manuscript to address the specific points below.

Reply: We sincerely appreciate the careful review and feedback supplied by the reviewer.

Comment: 1) The sentence on p. 8, lines 244-246 contains an error. “The side chain of K745, the catalytic lysine, exhibits a “swing” toward the benzo ketone in the case of 2 binding opening a position on the imidazole, which now binds a solvent water (Figure 2D).” Obviously, a comma is missing between “binding” and “opening”. Please correct.

Reply: We have corrected this typo.

Comment: 2. The reference “Figure 1 C-D” at the end of the sentence on p. 11, lines 299-301 is incorrect. There is no Figure 1D as referenced because obviously Figure 2 C-D is meant. Please change accordingly.

Reply: We have corrected this typo.

Comment: 3. To improve the clarity of the structural figures in general, the colors and labeling of the individual compounds/inhibitors should be consistent, at least per figure. Nearly all structural figures (X-ray, MD simulations, and docking) show both compounds in subfigures and are not consistently labeled. It is time-consuming to look up which compound is which in the figure legend. I would suggest that each subfigure containing structures is consistently labeled with the compound number in the color of the compound or in black throughout the manuscript and supplementary information. For example, in Figure 2c and d both compounds are colored in green and are labeled with black numbers. In Figure 2e, the coloring of compound 1 changed to yellow, while compound 2 is still green but now labeled with a green number. In Figure S2 compounds are not labeled at all and in the figure legend for Figure S2f obviously, the wrong compound number is noted (for details see point 5 below). Likewise, in Figure S7 and S8 compounds are not labeled. To solve the problem, I propose to reserve one color each for compounds 1 and 2 and use this coloring throughout the manuscript, but at least it should be consistent in one figure.

Reply: We thank the reviewer for the critique regarding the structural figures and have taken the opportunity to enhance the consistency of the images throughout the manuscript and supplemental information. We have edited the images featuring **1**, **2**, and **DDC4002** to consistently showcase these molecules as yellow, green, and magenta, respectively. Exceptions to this include the B-factor visualization in Fig 3A-B. All compound labels are edited to be black.

Comment:4. By calculating a simulated-annealing Fo-Fc omit map, the authors show unambiguously that compounds 1 or 2 are bound to the ATP- and allosteric site of the protein. However, it is necessary that the sigma levels of all electron density map snapshots are written in the legend of Figure S2. Furthermore, a consistent coloring of 2Fo-Fc and Fo-Fc electron density maps would improve the clarity of Figures S2B, C, E, and F. Currently, both 2Fo-Fc maps are colored in blue, while the Fo-Fc map for compound 1 is in purple and the one for compound 2 is again in blue, which is quite

confusing. The coloring should generally be consistent and explained in the figure legend, especially if different map types and structures with different compounds are shown side by side.

Reply: We have updated the Figure caption to Fig S2 to include the sigma levels visualized, have consistent colors of the density maps, and feature ligands visualized in colors consistent with the manuscript figures.

Comment: 5. The figure legend of Figure S2f (p. S4, line 88) contains an error. It states that the Fo-Fc map of compound 1 is shown, however, it actually shows compound 2 with its omit map.

Reply: We have fixed this typo.

Comment:6. The legend of Figure S2 (p. S4, lines 83-88) does not contain a figure title, like all the other figures. Obviously, the title of the legend was accidentally shifted. I assume that the sentence "Electron density maps for cocrystal structures of 1 and 2 in complex with EGFR(T790M/V948R).", which is misplaced in the middle of the legend, actually is the title of the figure and should be moved to the beginning.

Reply: We have fixed this typo.

Comment: 7. The crystallographic work is generally technically sound but the crystallographic table has to be improved. A lot of physical units in Table S2 reporting on the crystallographic data statistics are missing. Please add missing units for the wavelength [\AA], the resolution range [\AA], the unit cell dimensions [\AA] and angles [$^\circ$], the Wilson B-factor [\AA^2], the RMSD bonds [\AA] and angles [$^\circ$] as well as the average B-factors [\AA^2]. Furthermore, the units for the B-factor [\AA^2] are missing in Figure 3a and b, Supplementary Figure S8a and b as well as in their figure legends (p. 13, lines 332-333 and p. S11, lines 156-157).

Reply: Thank you for noting these omissions. We have added units to crystallographic tables and figures and captions across the manuscript as mentioned by the reviewer.

Comment: 8. The legend of Figure S3 (p. S5, lines 90-92) is incomplete. Please mention the colors of the compounds and the color of the activation loop in the figure legend.

Reply: We have corrected this omission.

Comment: 9. Tables S3 and S4 in supplementary information contain a mixture of different font types. Since supplementary information is usually not copy-edited please unify font types.

Reply: We have adjusted the fonts in tables S3 and S4 for consistency.

Comment: 10: Models derived from crystal structures are relaxed or equilibrated for some nanoseconds prior to the individual MD simulations, which is also mentioned in the methods section. For how long have the models been relaxed?

Reply: Relaxation was performed according to Desmond's standard protocol for energy minimization, namely 12ps for NVT, followed by 12ps NPT.

Comment: 11. The RMSDs especially for the protein in the MD simulations (Figure S6) sometimes "suddenly jump" by $> 2 \text{ \AA}$ (e.g. replica 2, 3, and 6 for compound 1 as well as replica 2 and 5 for compound 2). Could the authors explain what the reasons are for these large jumps when they check the models at these time points?

Reply: Those represent conformational changes that took place within the activation loop, which were not reflected on the ligand stability. Since overall simulation energy and interactions were taken from the means and average of trajectories, we believe the simulations were sufficiently long to disregard those effects as statistical outliers.

Comment: 12. Figure S7 shows the result of docking compound 4 to the EGFR kinase domain. What are the yellow dashed lines that reach to the amino acid side chains or the compounds on one end but go nowhere on the other end?

Reply: We have simplified Figure S7 to emphasize the binding mode of **4** compared to the experimentally-determined cocrystal structures of **2** and **7**.

Comment: One note on the information content of figures and supplementary figures that the authors may consider: Supplementary figures sometimes show the same information as the main figures. For example, Figure 3a and b (B-factor coloring of the ligand) shows the same information as Figure S8a and b. The only difference is the presence of the protein ribbon in Figure S8, which is, however, not discussed, mentioned in the text, or important otherwise.

Reply: We have added a sentence on page 11 highlighting the utility of showing the ligand B-factors in context of the protein ribbon for clarity.

Comment: Another example is Figure 2e showing compounds 1 and 2 superposed, and Figure S4c with exactly the same image but with an added angle and distance.

Reply: We have added the angle and distance to Figure 2 and deleted the corresponding panel C in Figure S4

REVIEWERS' COMMENTS:

Reviewer #2 (Remarks to the Author):

No more comments, publish as the revised manuscript.

Reviewer #3 (Remarks to the Author):

Thanks to all the authors for addressing point by point my suggestions.
I have really appreciated this work.

Reviewer #4 (Remarks to the Author):

The authors have carefully addressed all reviewer's comments. I recommend publication of the manuscript without further revisions.